# EGFRvIII Confers Sensitivity to Saracatinib in a STAT5-Dependent Manner in Glioblastoma

**DOI:** 10.3390/ijms25116279

**Published:** 2024-06-06

**Authors:** Mylan R. Blomquist, Ryan Eghlimi, Angad Beniwal, Dustin Grief, David G. Nascari, Landon Inge, Christopher P. Sereduk, Serdar Tuncali, Alison Roos, Hannah Inforzato, Ritin Sharma, Patrick Pirrotte, Shwetal Mehta, Shannon P. Fortin Ensign, Joseph C. Loftus, Nhan L. Tran

**Affiliations:** 1Mayo Clinic Alix School of Medicine, Mayo Clinic Arizona, Phoenix, AZ 85054, USA; blomquist.mylan@mayo.edu (M.R.B.); nascari.david@mayo.edu (D.G.N.); 2Department of Cancer Biology, Mayo Clinic Arizona, Scottsdale, AZ 85259, USAtuncalisecz@gmail.com (S.T.); aroos29@gmail.com (A.R.); hinforza@asu.edu (H.I.);; 3Ventana Medical Systems, Roche Diagnostics, Tucson, AZ 85755, USA; 4Collaborative Center for Translational Mass Spectrometry, The Translational Genomics Research Institute, Phoenix, AZ 85004, USA; rsharma@tgen.org (R.S.);; 5Department of Translational Neuroscience, Barrow Neurological Institute, St. Joseph’s Hospital and Medical Center, Phoenix, AZ 85013, USA; 6Department of Hematology and Oncology, Mayo Clinic Arizona, Phoenix, AZ 85054, USA; 7Department of Neurological Surgery, Mayo Clinic Arizona, Phoenix, AZ 85013, USA

**Keywords:** glioblastoma, cell signaling, receptor tyrosine kinase, precision oncology

## Abstract

Glioblastoma (GBM) is the most common primary malignant brain tumor in adults, with few effective treatments. EGFR alterations, including expression of the truncated variant EGFRvIII, are among the most frequent genomic changes in these tumors. EGFRvIII is known to preferentially signal through STAT5 for oncogenic activation in GBM, yet targeting EGFRvIII has yielded limited clinical success to date. In this study, we employed patient-derived xenograft (PDX) models expressing EGFRvIII to determine the key points of therapeutic vulnerability within the EGFRvIII-STAT5 signaling axis in GBM. Our findings reveal that exogenous expression of paralogs STAT5A and STAT5B augments cell proliferation and that inhibition of STAT5 phosphorylation in vivo improves overall survival in combination with temozolomide (TMZ). STAT5 phosphorylation is independent of JAK1 and JAK2 signaling, instead requiring Src family kinase (SFK) activity. Saracatinib, an SFK inhibitor, attenuates phosphorylation of STAT5 and preferentially sensitizes EGFRvIII+ GBM cells to undergo apoptotic cell death relative to wild-type EGFR. Constitutively active STAT5A or STAT5B mitigates saracatinib sensitivity in EGFRvIII+ cells. In vivo, saracatinib treatment decreased survival in mice bearing EGFR WT tumors compared to the control, yet in EGFRvIII+ tumors, treatment with saracatinib in combination with TMZ preferentially improves survival.

## 1. Statement of Implication

STAT5 is a key therapeutic target specific for EGFRvIII+ GBM. Targeting the EGFRvIII-STAT5 axis with Src inhibition represents a specific clinically relevant vulnerability within this pathway and informs patient selection for precision medicine efforts to improve survival in glioblastoma.

## 2. Introduction

Glioblastoma (GBM), the most common primary malignant brain tumor in adults, confers a dismal ~15-month median survival at new diagnosis. The current standard of care for primary GBM includes maximally safe resection, radiation therapy, and concomitant and adjuvant chemotherapy with the DNA alkylating agent TMZ (TMZ) [1]. Despite these measures, GBM tumors inevitably recur due to diffuse tumor cell invasion throughout the brain parenchyma. These invasive cells escape surgical resection and reside in areas where the blood-brain barrier remains intact, hindering their pharmacologic accessibility [2]. Recurrent GBM tumors arising from these cells are inherently invasive, drug resistant, and exceptionally difficult to treat [3]. Recurrent GBM exhibits extensive genetic and phenotypic heterogeneity along with considerable cell-state plasticity, rendering inhibition of single drivers ineffective. A study comparing the genomic and transcriptomic profile of primary and recurrent GBM tumors by our group demonstrates that the activation of certain signaling pathways, rather than any specific genetic lesion, is maintained across primary and recurrent GBM [4]. Thus, the identification and inhibition of signaling pathways rather than targeting an individual driver alteration may serve as an effective strategy in GBM.

EGFR is an important oncogenic driver of GBM; EGFR amplification occurs in approximately 40–60% of primary GBMs, resulting in aberrant ligand-independent growth factor activation. Half of EGFR-amplified tumors also express EGFRvIII, a truncated exon variant inducing low-level constitutive EGFR activity. While EGFR inhibition provides survival benefit in carcinomas expressing EGFR variants in residues of the kinase domain, these inhibitors have not been successful in GBM. In the absence of effective EGFR inhibitors in GBM, the investigation of downstream pathways, including signal transducer and activator of transcription (STAT) family transcription factors (TFs), becomes necessary to disrupt this oncogenic signaling axis in GBM. The STAT TFs became attractive therapeutic targets in cancer with the increased recognition of transcriptional addiction as a mechanism of oncogenesis [5]. While rarely mutated, STAT TFs promote carcinogenesis in many malignancies, including GBM [6]. STAT family TFs are canonically activated by Janus kinases (JAK) 1 and/or 2 downstream of more than 50 cytokines and growth receptors on the cell surface, including epidermal growth factor receptor (EGFR).

Once activated via tyrosine phosphorylation, cytoplasmic STAT TFs dimerize, translocate to the nucleus, bind DNA, and induce transcription of target genes [7], inducing a range of phenotypic features involved in growth, proliferation, and differentiation [8]. In GBM, STAT3 acts as an oncogenic transcription factor [9], modulator of the immune microenvironment [10], and proposed marker for stratifying patients for targeted therapy [11]. Efforts to clinically inhibit STAT3 are underway, including a phase I clinical trial of a small-molecule inhibitor targeting JAK2 activity, WP1066, to suppress STAT3 function in recurrent malignant glioma [12]. 

Despite abundant studies of STAT3, the role of STAT5 and its potential as a therapeutic target in GBM remain understudied thus far. Our group previously reported that STAT5 is active in glioma cells residing at the invasive front of the tumor and promotes cell invasion and survival in GBM [13]. STAT5 is activated by phosphorylation of EGFRvIII by wild-type EGFR [14]. STAT5 encompasses protein products from two different genes, STAT5A and STAT5B [15]. While STAT5B expression score has been associated with poor outcome in GBM patients in one study [16], it is not clear if STAT5A plays an independent, analogous, or divergent role. A previous report in established glioma cell lines shows that EGFRvIII-induced activation of STAT5B through Src family kinase Fyn induces expression of BCL-XL and contributes to resistance to cisplatin [16]. In this study, we examine the role of both STAT5A and STAT5B in proliferation and sensitivity to STAT5 inhibition downstream of EGFRvIII signaling.

We demonstrate that oncogenic STAT5 signaling is a viable target in GBM. Corroborating other studies [16,17], we identify that STAT5 activation downstream of EGFRvIII is independent of canonical JAK1/2 activation, instead depending upon Src family kinase (SFK) activation. Saracatinib, a broad-acting EGFR/SFK inhibitor that crosses the blood-brain barrier, preferentially induces cell death in EGFRvIII-expressing cells, which is rescued by the introduction of either constitutively active STAT5A or STAT5B. We found that saracatinib sensitizes EGFRvIII+ tumors to TMZ in a model of orthotopic GBM implanted in nude mice. Cumulatively, these results indicate that EGFRvIII status may serve as a biomarker for clinical saracatinib use and that the cell death induced by saracatinib in the EGFRvIII condition is primarily due to STAT5 inhibition. These preclinical results demonstrate that the EGFRvIII/Src/STAT5 axis merits further study as a translationally relevant signaling pathway in GBM.

## 3. Results

### 3.1. STAT5 Activation Is Induced Downstream of EGFRvIII, Independent of EGF Ligand

While phosphorylation of STAT3 remains relatively consistent across PDX lines of GBM independent of EGFR status, phosphorylation of STAT5 varies significantly across PDX lines established from 10 different patients (Figure 1A). We queried matched sets of lysates from parental PDX tumors and corresponding TMZ-resistant tumors generated in a previous study [18]. In four cases (GBM6, GBM10, GBM12, and GBM39), phosphorylation of STAT5 increased following in vivo induction of TMZ resistance (R). All EGFRvIII-bearing tumors expressed phosphorylated STAT5, either in the primary PDX or in the matched TMZ-resistant PDX. 

There are limited data on STAT5 in GBM, likely owing to the loss of key oncogenic drivers, such as EGFR amplification and/or EGFRvIII expression, in long-term established culture models, resulting in decreased STAT5 activation [19]. We examined STAT5 and STAT3 phosphorylation at the canonical activation sites (STAT5A Y694/STAT5B 699 and STAT3 Y705) in LN229 and GBM22 stably expressing either wild-type EGFR, EGFRvIII, or co-expressing wild-type EGFR and EGFRvIII (referred to as EGFR/EGFRvIII). Under serum-deprived conditions, STAT3 phosphorylation is maintained at Y705 regardless of EGFR status. However, STAT5 Y694/Y699 phosphorylation in the absence of ligand is dependent on EGFRvIII (Figure 1B), corroborating previous reports [20]. To assess whether EGFRvIII and STAT5 form a stable complex in our models, we utilized an antibody against the cytoplasmic domain of EGFR to capture both full-length EGFR and EGFRvIII via immunoprecipitation in GBM22 cells expressing either WT EGFR or EGFR/EGFRvIII (Figure 1C). STAT3 preferentially co-immunoprecipitates with EGFR in the WT EGFR condition, whereas STAT5 preferentially co-immunoprecipitates with EGFR/EGFRvIII (Figure 1C). 

### 3.2. STAT5A and STAT5B Both Augment Proliferation in GBM PDX Models

To examine the roles of each STAT5 paralog in GBM proliferation, expression of STAT5A and STAT5B was stably induced in two PDX cell lines, GBM22 and GBM39 (Figure 1D). Figure 1D indicates the expression level of STAT5A (HA-tagged) and STAT5B (FLAG-tagged) in each cell line. Expression of STAT5A or STAT5B results in an increased rate of proliferation, significantly decreasing the doubling time in the logarithmic phase of both cell lines and the time to reach confluency (Figure 1E, Appendix A). There is no significant difference in proliferation induced by exogenous expression of STAT5A compared to STAT5B. 

### 3.3. Pharmacologic Inhibition of STAT5 Concomitant with TMZ Prolongs Survival In Vivo

Pimozide, a first-generation antipsychotic drug used to treat Tourette syndrome and treatment-resistant tics, has been shown to inhibit STAT5 in myelogenous leukemia [21], osteosarcoma [22], and GBM [13] cells in vitro. Given the preferential activation of STAT5 in EGFRvIII-expressing cells and its role in GBM proliferation, we used pimozide to inhibit STAT5 in mice orthotopically implanted with GBM6, an EGFRvIII+ patient-derived xenograft. Mice in the combination group (PIM + TMZ) were pre-treated with 10 mg/kg pimozide daily for 1 week prior to TMZ chemotherapy. The following weeks, mice were treated daily with pimozide (am) and TMZ (pm). While pimozide alone does not provide survival benefit, pimozide combined with TMZ, the standard chemotherapeutic used in GBM, prolonged animal survival significantly compared to TMZ alone (Figure 2A). Tyrosine phosphorylation of STAT5 and STAT5 transcriptional target Fn14 were successfully inhibited in the orthotopic GBM6 tumors with 10 mg/kg of pimozide (Figure 2B), supporting STAT5 as a viable therapeutic target in EGFRvIII+ GBM. However, while combination of TMZ with STAT5 inhibition prolongs survival in EGFRvIII+ GBM, we calculated the equivalent dose of pimozide required to inhibit STAT5 in patient CNS tumors (0.8 mg/kg in humans) and found this to be above the safe maximum dose of no more than 0.2 mg/kg in adults, requiring alternative translational and clinical approaches to EGFRvIII-STAT5 signaling inhibition.

### 3.4. Src Family Kinase Activity Is Required for EGFRvIII/STAT5 Association and STAT5 Phosphorylation

To identify additional points of therapeutic vulnerability along the EGFRvIII-STAT5 signaling axis, we next determine whether STAT5 is activated canonically through JAK family members downstream of EGFRvIII. STAT3 activation (p-Y705) in the ligand-independent EGFR/EGFRvIII context requires JAK1, while phosphorylation of STAT5 (Y694/699) is independent of JAK1, JAK2 (Appendix A), and TYK2 (Appendix A). 

Next, we sought candidate kinases that are preferentially active in the EGFRvIII+ GBM compared to WT EGFR. An activity-based proteomic profiling (ABPP) assay of isogenic GBM22 cells identified kinase activity unique to the EGFRvIII condition compared to WT EGFR alone, including the receptor tyrosine kinases (RTK) MET and AXL and cytoplasmic non-receptor tyrosine kinases Src, CHK2, PKCa, MAP2K3, MAP2K2, and others (indicated as red dots, Figure 3A). Because AXL and MET have been shown to become transactivated by RTK heterodimerization with EGFR and EGFRvIII [23,24] and to initiate compensatory signaling pathways in the context of resistance to EGFR inhibitors in several cancers, including GBM [25,26,27,28], we assessed the potential contribution of these RTKs to STAT5 phosphorylation. STAT5 phosphorylation is unchanged in the context of AXL or MET knockdown (Appendix A), indicating that the activation of STAT5 under EGFRvIII is independent of RTK crosstalk with AXL or MET.

Given our ABPP results and prior reports that Src-dependent STAT activation occurs downstream of EGFRvIII in established cell lines [16,17], we focused on Src family kinase inhibition as a means of disrupting the EGFRvIII-STAT5 axis. Broad-acting Src family kinase inhibitors saracatinib and dasatinib both inhibit STAT5 phosphorylation under serum-starved conditions in LN229 cells expressing EGFR/EGFRvIII (Figure 3B). Both inhibitors also decrease STAT5 phosphorylation induced by the addition of an EGF ligand (Figure 3B), indicating that ligand-dependent signaling through WT EGFR cannot overcome inhibition of STAT5 phosphorylation by dasatinib and saracatinib. STAT3 phosphorylation is not disrupted by either inhibitor. Saracatinib depletes constitutive phosphorylation of STAT5 in a dose-dependent manner in GBM39, a PDX cell line endogenously expressing EGFRvIII with constitutive STAT5 activation (Figure 3C). To assess whether STAT5 complexing with EGFRvIII is dependent upon Src signaling, we treated GBM39 cells with multiple SFK inhibitors and performed immunoprecipitation to evaluate EGFRvIII/STAT5 complex formation (Figure 3D). Both dasatinib and saracatinib showed disruption of the EGFRvIII/STAT5 complex. Bafetinib, an inhibitor of BCR/Abl and Src family kinase member LYN, attenuated but did not eliminate EGFRvIII/STAT5 complex formation (Figure 3D), suggesting that activated LYN may participate in complex formation but is not solely required for EGFRvIII/STAT5 complex formation.

### 3.5. Apoptosis Induced by Saracatinib in EGFRvIII+ GBM Cells Is Dependent on STAT5A/B Paralogs

We sought to determine whether EGFRvIII sensitizes cells to saracatinib. GBM22 cells expressing EGFRvIII and GBM39, which harbors endogenous EGFRvIII, were treated with saracatinib and evaluated for apoptotic cell death measured by cleaved PARP. EGFRvIII+ GBM cells displayed increased cleaved PARP compared to GBM22 EGFR WT or parental GBM22 (Figure 4A), as well as increased accumulation of annexin V when treated with saracatinib and TMZ (Figure 4B,C). We next evaluated whether saracatinib sensitivity in EGFRvIII+ GBM cells is dependent specifically on either STAT5 paralog. Expression of either constitutively active STAT5A or STAT5B rescued saracatinib-induced apoptosis in EGFR/EGFRvIII-expressing GBM cells (Figure 4B,C, bottom panels). These data support the fact that the preferential sensitivity of EGFRvIII+ cells to saracatinib is dependent upon STAT5 and can be attributed to contributions from STAT5A and STAT5B activity downstream of the EGFRvIII/Src signaling axis. Thus, saracatinib may represent a viable means of targeting STAT5 through EGFRvIII/Src axis inhibition.

### 3.6. Saracatinib Sensitizes EGFRvIII+ GBM Cells to TMZ and Increases Survival of EGFRvIII+ Tumor-Bearing Mice

Our in vitro data demonstrate that EGFR/EGFRvIII tumor cells are more sensitive to saracatinib than EGFR WT tumor cells. To test this hypothesis in vivo, we used an orthotopic mouse model of GBM, implanting GBM22 cells expressing EGFR WT or EGFR/EGFRvIII and luciferase into athymic nude mice. The presence of EGFRvIII significantly reduced animal survival compared to EGFR WT in untreated controls (Figure 5A). Unexpectedly, saracatinib treatment significantly decreased survival in the GBM22 EGFR WT condition (Figure 5A). This impact of saracatinib monotherapy on survival time is not observed in the EGFR/EGFRvIII group. We explored possible explanations for this effect of saracatinib in EGFR WT tumors and noted that there was no consistent increase in tumor size at animal sacrifice from those in other treatment groups (Appendix A). However, GBM22 EGFR WT tumors treated with saracatinib displayed an increase in invasive phenotype (Appendix A), with extensive invasion beyond the tumor boundary. GBM22 EGFR/EGFRvIII tumors treated with saracatinib were relatively well-circumscribed by comparison (Appendix A).

Next, we explored the combination of saracatinib and TMZ in GBM22 EGFR WT and EGFR/EGFRvIII tumors. The addition of saracatinib to TMZ provided a significant survival benefit compared to TMZ alone in EGFR/EGFRvIII tumors (Figure 5B). No additional survival benefit was noted in EGFR WT tumor-bearing mice treated with saracatinib and TMZ (Appendix A).

## 4. Discussion

Our results indicate that the EGFRvIII/STAT5 signaling is a targetable axis in GBM with unique therapeutic vulnerabilities. While STAT3 has been extensively studied in solid tumors, including GBM, less is known about the role of STAT5A and STAT5B in tumor progression. We find that the introduction of exogenous STAT5A and STAT5B individually augments proliferation in GBM cell lines, demonstrating that both STAT5 paralogs may independently contribute to tumor phenotype. Previous studies favor STAT5B as the primary oncogenic STAT5 paralog [16,29], but our study supports further investigation of STAT5A and whether these paralogs exhibit redundant or complementary roles in GBM phenotype. Here, we validated that inhibition of STAT5 in EGFRvIII+ GBM is a clinically relevant strategy in orthotopic PDX tumor models in vivo. STAT5 inhibition with pimozide is limited to use as a tool compound. Pimozide acts non-specifically, and the mechanism of STAT5 inhibition by pimozide is not clear [21]. Further, the doses used in this study to achieve improved survival paired with TMZ (10 mg/kg pimozide in mice) exceed the safe maximum dose in humans, carrying the risk of ventricular arrhythmia. Our data argue for the development of more specific STAT5 inhibitors for EGFRvIII+ GBM or the identification of an alternative pharmacologic approach to inhibit the EGFRvIII-STAT5 signaling axis.

We thus explored the EGFRvIII-STAT5 signaling axis to uncover further potential therapeutic vulnerability. In our model systems, the activation of STAT5 downstream of EGFRvIII was independent of Janus kinases, including JAK1, JAK2, and TYK2, instead requiring Src family kinase activity. Several Src family kinases are known to be expressed in GBM, including Src, Lyn, Fyn, Yes, and Lck [30]. Although we have yet to identify a Src family kinase member consistently required for phosphorylation of STAT5 in each of our cell models (LN229, GBM22, and GBM39), it is most likely that a combination of Src family kinases is responsible for STAT5 activation and that this combination varies from tumor to tumor. Despite the potential heterogeneity of Src family kinase-mediated activation of STAT5 in GBM [31,32,33], we validated that inhibition with non-specific Src family kinase inhibitors, such as saracatinib and dasatinib, represents a broadly applicable means of inhibiting the EGFRvIII-Src-STAT5 axis and reliably inhibits phosphorylation of STAT5. 

Importantly, our data highlight the critical role of patient tumor stratification by EGFR status, as saracatinib represents a promising therapeutic strategy that we find specific to EGFRvIII+ tumors. We show that saracatinib, paired with TMZ, provides a survival benefit in PDX cell line GBM22 expressing EGFR/EGFRvIII, which was not seen in EGFR WT expressing tumors. Saracatinib demonstrates blood-brain barrier penetrance and a favorable pharmacokinetic profile, with sustained inhibition of the target in vivo in a murine Alzheimer disease model [34]. In patients, saracatinib was well tolerated at doses up to 175 mg per day and found to inhibit Src activity in advanced solid tumors in a phase I clinical trial [35]. While there have not yet been any clinical trials of saracatinib in any CNS malignancy, a phase II clinical trial of dasatinib monotherapy in recurrent GBM tumors with at least two dasatinib tyrosine kinase targets activated or overexpressed did not provide clinical benefit [36]. A randomized phase II trial of dasatinib combined with chemotherapy and RT also did not provide survival benefit to GBM patients (NCT00869401) [37]. Our study suggests that instead of Src activation, the presence of EGFRvIII may serve as a more effective biomarker to stratify for sensitivity to saracatinib, especially given the varied activation of different Src family kinases between individual tumors and dependence on downstream mediators, including STAT5A and STAT5B, in exerting phenotype. 

Saracatinib treatment resulted in shorter survival compared to the untreated controls only in mice bearing EGFR WT tumors. While the tumors were not appreciably larger in the EGFR WT saracatinib-only treated group compared to no treatment control, we found that cells from these tumors invaded beyond the boundary of the tumor into the brain parenchyma to a greater degree than control tumors. One possible explanation for these results is that the EGFR WT tumors, but not those bearing EGFRvIII, were driven to become more invasive by saracatinib treatment. This effect is counterintuitive given the importance of Src in pro-invasive signaling, including through the MAPK pathway [38] and focal adhesion kinase (FAK) [39]. The impact of saracatinib on the phenotype of EGFR-WT-expressing GBM tumors requires further investigation. 

In our model, GBM22 EGFR/EGFRvIII tumors were more resistant to TMZ than GBM22 EGFR WT. Animals harboring GBM22 EGFR WT tumors survived longer than GBM22 EGFR/EGFRvIII tumors when treated with TMZ monotherapy, suggesting the addition of saracatinib preferentially improves response in GBM cells with more TMZ resistance. Our model supports that EGFRvIII+ GBM is associated with increased TMZ resistance. It is possible that the combination of saracatinib and TMZ in EGFRvIII+ GBM cells provides a unique therapeutic strategy to target more TMZ-resistant tumor populations. Our study highlights the critical importance of patient selection in precision medicine in oncology. We identified that EGFRvIII status in GBM holds potential in predicting responders to saracatinib and anticipate that future studies will be able to determine the best combination of cytotoxic or targeted drugs to pair with saracatinib to optimally treat EGFRvIII+ GBM.

## 5. Materials and Methods

### 5.1. Antibodies and Reagents

Antibodies against phosphorylated EGFR (Y1068, #3777), total EGFR (#2239), phosphorylated STAT3 (Y705, #9145), total STAT3 (#9139), phosphorylated STAT5 (Y694/699, #9359), STAT5A (#4807), FLAG tag (#14793), HA tag (#3724), phosphorylated AKT (#4060), pan-AKT(#2920), JAK1 (#50996), JAK2 (#3230), TYK2 (#14193), GAPDH (#5174), cleaved PARP (#5625), phosphorylated AXL (#96453), total AXL (#8661), phosphorylated MET (#3126) and MET (#3127) were purchased from Cell Signaling Technology (Danvers, MA, USA). The anti-EGFR used for immunoprecipitation was purchased from Abcam (#52894), and the IgG isotype control used was #3900. Total STAT5B was purchased from Santa Cruz, CA, USA (sc1656). The anti-alpha-tubulin antibody is from Millipore, Burlington, MA, USA, (#05-829). Anti-Fn14 used in immunohistochemistry was purchased from R&D Systems, McKinley Place, MN, USA, (Mouse TWEAK R/TNFRSF12 antibody, #AF1610). Anti-Ki67 used in immunohistochemistry is from Abcam (ab16667). Recombinant EGF was purchased from PeproTech, Waltham, MA, USA, with working stock resuspended in 0.1% BSA/PBS. Annexin V dye for use with the Incucyte Live Cell Imaging System was purchased from Sartorius (#4641). Saracatinib (S1006) and dasatinib (S1021) were purchased from Selleck Chemicals, Houston, TX, USA. Pimozide (P1793) and TMZ (T2577) were purchased from Sigma-Aldrich (St. Louis, MO, USA). For in vitro experiments, all drugs were resuspended in DMSO except for TMZ, which was made in 0.1 M HCl.

### 5.2. Cell Culture

The primary GBM patient-derived xenograft lines GBM39 and GBM22 were acquired from the Mayo Clinic Brain Tumor Patient-Derived Xenograft National Resource, Phoenix, AZ, USA, [40]. Briefly, these cell lines were established from a patient surgical sample and maintained as a flank xenograft in immunodeficient mice. TMZ-resistant PDX cell lines were generated in vivo as described in [18]. GBM39 and GBM22 flank tumors were resected, brought to single-cell suspension via mechanical dissociation, and then subsequently maintained as adherent cultures in Dulbecco’s modified Eagle medium (DMEM) supplemented with 10% fetal bovine serum (FBS) for the studies herein unless alternative serum conditions are specified in the figure legends. LN229 and HEK293FT cells were purchased from the American Type Culture Collection, Manassas, VA, USA. All cells were kept in 1% penicillin-streptomycin for passage, but antibiotics were omitted for the duration of experiments. Cultures were routinely tested for mycoplasma contamination with the MycoAlert Detection Kit (Lonza, Basel, Switzerland). 

### 5.3. Expression Constructs and Generation of Isogenic Cell Lines

LN229 cells expressing EGFR or EGFR/EGFRvIII were a kind gift from Dr. Qi-Wen Fan, generated as described in [14]. GBM22 cells expressing EGFR or EGFR/EGFRvIII were generated from pWLZ-hygro-EGFR-as3 (a gift from Dr. Qi-Wen Fan) and pLRNL-neo-EGFRvIIIDY5 (a gift from Dr. Frank Furnari). 

Bacterial plasmids containing the coding sequence of either human STAT5A (clone ID: HsCD00043806) or human STAT5B (clone ID: CD00852071) were obtained from the DNASU plasmid repository (The Biodesign Institute/Arizona State University, Tempe, AZ, USA). The coding sequence for STAT5A with a C-terminal 3X HA epitope was amplified by PCR, and the purified fragment was subcloned into the lentiviral transfer vector pCDH-CMV-MCS-EF1a-GFP/Puro (System Biosciences, Palo Alto, CA, USA; catalog number CD513B-1). A constitutively active STAT5A (STAT5A-CA) containing the point mutation N642H [41] was generated using the QuickChange II Site Directed Mutagenesis kit (Agilent, Santa Clara, CA, USA). The coding sequence for STAT5B with a N-terminal 3× Flag epitope was amplified by PCR, and the purified fragment was subcloned into the lentiviral transfer vector pCDH-CMV-MCS-EF1a-GFP/Puro. As with STAT5A, the gain of function mutation N642H was introduced into the coding sequence of STAT5B using the QuickChange II Site Directed Mutagenesis kit (Agilent, Santa Clara, CA, USA). All mutations were verified by DNA sequencing. 

For production of lentivirus, the packaging cell line HEK293FT was co-transfected with 42 ug of transfer vector and the Horizon Discovery Trans-Lentiviral Packaging System (#TLP4606). HEK293FT cells were plated the day before co-transfection on a poly-d-lysine (Sigma, St. Louis, MO, USA; P6407) coated plate. Plasmid and Trans-Lentiviral Packaging Mix were combined with calcium chloride and 2× Hank’s Balanced Saline Solution (Horizon Discovery, Cambridge, UK) and added to cells overnight. Calcium-phosphate-containing media was removed and replaced with DMEM + 5% FBS and incubated for 48 h. Lentiviral supernatant was collected, sterile filtered using SteriFlip tubes (Millipore, Burlington, MA, USA), and precipitated with cold PEG-It Virus Precipitation Solution (System Biosciences, Palo Alto, CA, USA) overnight. The supernatant and PEG-It mixture were spun down and resuspended in cold, sterile phosphate-buffered saline and aliquoted. The TransDux MAX Lentivirus Transduction Reagent (System Biosciences, Palo Alto, CA, USA) was used for lentiviral transduction of target cells. TransDux and MAX enhancers were used at concentrations suggested by the manufacturer. Antibiotic selection was initiated 48–72 h following lentiviral transduction, with either 1 ug/mL of puromycin or 200 ug/mL hygromycin, depending on sensitivity, for two weeks. Stable expression was validated by Western blot and/or expression of green or red fluorescent protein prior to experiments.

### 5.4. Transfection with Small Interfering RNA (siRNA)

JAK1 and JAK2 were transiently inhibited with Qiagen FlexiTube siRNA, Hilden, Germany. TYK2 was transiently inhibited with Dharmacon ON-TARGET Plus siRNA. Each of the four siRNA sequences, against the same target, included in the Flexitube siRNA kit or the Dharmacon ON-TARGET Plus kit were tested for knockdown by Western blot. The two most complete knockdowns for each target out of the four sequences, listed below, were chosen for experiments. Transient transfection of siRNA at 10 nM was achieved using Lipofectamine RNAiMax reagent according to the manufacturer’s protocol presented in Table 1.

### 5.5. Immunohistochemistry

Brains were harvested at the time of sacrifice and fixed overnight in 10% neutral-buffered formalin and embedded in paraffin. Formalin-fixed, paraffin-embedded (FFPE) samples were sectioned, placed onto charged slides, and baked for one hour. For Figure 2B, five-micron-thick slides were subject to IHC staining using the Leica Bond RXm automated IHC stainer (Leica Biosystems, Deer Park, IL, USA), with protocol and antigen retrieval as indicated in [13]. Quantification of cells expressing Fn14 or phosphorylated STAT5 was determined using inForm Tissue Analysis Software version R 3.1.0 (Akoya Biosciences, Marlborough, MA, USA).

For Appendix A, five-micron-thick slides were deparaffinized and subjected to citrate-based antigen retrieval (Vector Antigen Unmasking Solution, Citrate-Based (H-3300-250)) in a pressure cooker. Washes were completed in TBST, and blocking and antibody dilution were achieved with 5% goat serum in TBST. Ki67 antibody was diluted at 1:1000. Slides were mounted in CytoSeal 60 (Electron Microscopy Sciences, Hatfield, PA, USA). Slide scans acquired with Akoya Vectra 3. H&E stain was performed using standardized methods.

### 5.6. Immunoblotting and Immunoprecipitation

Monolayers of adherent cells were washed once with phosphate-buffered saline (PBS) containing 1 mmol/L phenylmethylsulfonylfluoride on ice and lysed in 1× RIPA buffer (50 mM Tris, pH 8.0, 135 mM NaCl, 1% NP-40, 0.1% SDS, 0.5% sodium deoxycholate, 5% glycerol) supplemented with 1× HALT protease and phosphatase inhibitor (Thermo Fisher Scientific, San Jose, CA, USA). Lysates were sonicated, and protein concentration was determined using BCA Assay (Thermo Fisher Scientific, San Jose, CA, USA Pierce) with bovine serum albumin as a standard. Forty micrograms of total protein were loaded per lane and separated via a 4–12% gradient or 10% SDS-PAGE gel using the Novex Surelock Mini electrophoresis unit. Protein was transferred to a nitrocellulose membrane using the BioRad fast transfer system. Membranes were briefly rinsed with diH2O and blocked for at least 30 min at room temperature with Odyssey Blocking Buffer (LI-COR). Primary and secondary antibodies were diluted in Odyssey Blocking Buffer with 0.1% Tween-20. Protein bands were then detected with the Odyssey CLx Near-Infrared (NIR) Western Blot Detection System (LI-COR Biosciences, Lincoln, NE, USA).

For immunoprecipitation, cells were lysed in Pierce buffer (Thermo Fisher Scientific, San Jose, CA, USA) with HALT protease and phosphatase inhibitor. Moreover, 1 mg of protein for each sample was pre-cleared with Dynabeads Protein G (Thermo Fisher Scientific, San Jose, CA, USA) for 30 min at 4 °C. Pre-cleared lysates were incubated with the primary antibody at 4 °C overnight. A mix of all lysates, approximately the same proportion from each sample, was incubated with the isotype control for 4 °C overnight. Antibody-bound protein was pulled down from whole cell lysate with a one-hour incubation with Dynabeads at room temperature. The beads were washed 3× in cold Pierce buffer, then resuspended in 3× blue loading buffer (Cell Signaling Technology, Danvers, MA, USA; #7722S), water, and reducing agent 30× dithiothreitol (DTT) and boiled. The beads were then magnetized and the immunoprecipitated separated by SDS-PAGE and immunoblotted as above. In instances in which the same lysates were run on more than one gel, separate alpha-tubulin controls are included for each gel.

### 5.7. Proliferation Assay and Doubling Time

Cell proliferation was measured by percent confluence, derived from live-cell images captured at 10× magnification on a Sartorius Incucyte SX5 Live-Cell Analysis Instrument. Moreover, 10,000 cells per well were seeded in 200 µL of 10% FBS/DMEM per well in a clear-bottom 96-well microplate and allowed to adhere to the plate for at least 4 h prior to initial image acquisition. Moreover, 12 wells per condition were plated unless otherwise specified. Following cell adhesion, images were taken every 6 h for approximately 7 days. Confluence was calculated using the Basic Analyzer function on the Incucyte SX5 Live-Cell Analysis System. Test images from days 1, 3, and 7 were used to determine the fitness of the analysis mask. Confluence values (in percent) were exported and plotted in GraphPad Prism.

Doubling time of cell lines in the logarithmic phase was determined by averaging all replicate confluence values at the initial 0 h read, *a*, and all replicate confluence values at 60 h, *b*. The confluence values at 60 h are estimated to fall within the logarithmic phase of proliferation of most cell lines used in this study. Doubling time was calculated by (60*ln(2))/(ln(*b/a*)). 

### 5.8. In Vivo Studies 

#### 5.8.1. Pimozide and TMZ (Figure 2)

The Mayo Clinic Institutional Animal Care and Use Committee approved the following experiments. Luciferized GBM6 (fLuc-GFP) was implanted into 40 athymic nude mice using stereotactic injection of 500,000 cells per mouse as previously described [42]. Mice were randomized three weeks later into the four treatment arms based on tumor size, with each treatment arm having an approximately equal average tumor size as determined by bioluminescence. Mice were first pre-treated with pimozide before the initiation of TMZ. At week 4 post-implantation, treatment was initiated with 10 mg/kg pimozide suspended in ORA-Plus (Padagis, Minneapolis, MN, USA) by oral gavage once per day (am) in pimozide-only and combined (TMZ plus pimozide) groups. At week 6 post-implantation, TMZ treatment was initiated, with mice in the TMZ-only and combination groups receiving 50 mg/kg oral TMZ suspended in ORA-Plus daily (pm). Mice were weighed 2–3 times per week and sacrificed upon losing more than or equal to 20% of their starting body weight or demonstrating any neurological symptoms. 

#### 5.8.2. Saracatinib and TMZ (Figure 5)

Luciferized GBM22 expressing either EGFR WT or EGFR/EGFRvIII were implanted into 40 athymic nude mice per cell line using stereotactic injection of 300,000 cells per mouse. The following week, mice were randomized into the four treatment arms based on tumor size, with each treatment arm having an approximately equal average tumor size as determined by bioluminescence. Doses of saracatinib and TMZ were administered via oral gavage in a 50:50 mix of ORA-Plus and ORA-Sweet (Padagis). Mice underwent three one-week cycles of 25 mg/kg saracatinib (SelleckChem #S1006) 5 days/week (am) and/or 50 mg/kg TMZ (compounded by Mayo Clinic pharmacy, Scottsdale, AZ, USA) 5 days/week (pm). Mice were weighed 2–3 times per week and sacrificed upon losing more than or equal to 20% of their starting body weight or demonstrating any neurological symptoms.

### 5.9. Activity-Based Protein Profiling 

ABPP experiments were carried out using Pierce^®^ Kinase Enrichment Kits and ActivX^®^ ATP Probes (ThermoFisher Scientific, San Jose, CA, USA), according to the manufacturers’ instructions and as previously published [43,44]. Briefly, GBM22 cell pellets from different conditions (parental, EGFR wt, EGFR vIII, and EGFR wt + vIII) were resuspended in 400 µL of lysis buffer and 4 µL of Halt TM phosphatase and protease inhibitor cocktail (Catalog #78440, ThermoFisher Scientfic, San Jose, CA, USA). Following sonication (3 cycles of 30 s at 1-min intervals using a pulse of 50% duty cycle on VCX130 Vibra-Cell TM, Sonics, Austin, TX, USA), lysates were cleared by centrifugation at 14,000× g for 20 min at 4 °C. Supernatants were buffer exchanged using Zeba Spin Desalting Columns (ThermoFisher Scientific, San Jose, CA, USA). Protein concentrations were estimated using BCA assays, and a total of 780 µg of protein from each lysate was prepared for labeling, enrichment, and LC-MS/MS analysis. Protein extracts were first incubated with 20 mM MgCl_2_, followed by incubation with 10 µM of DesThioBiotinylated (DBT)-ATP probes for 10 min. Following labeling, proteins were denatured in 10 M urea and reduced with 5 mM DTT. Samples were alkylated by incubation with 40 mM iodoacetamide in the dark for 30 min at room temperature. Following desalting (Zeba spin columns), proteins were digested using trypsin (1:50 enzyme-to-substrate ratio) overnight at 37 °C. DBT-labeled peptides were captured by incubating the digests with 50 µL of high-capacity streptavidin beads for 1 h. The beads were sequentially washed with lysis buffer, PBS, and HPLC-grade water; bound peptides were eluted by addition of aqueous 50% acetonitrile with 0.1% trifluoroacetic acid (TFA). The eluted peptides were vacuum-concentrated and resuspended in loading solvent (98% water, 2% acetonitrile, 0.1% formic acid) for LC-MS/MS analysis. 

### 5.10. Liquid Chromatography and Mass Spectrometry

LC-MS/MS data were acquired on a U3000 RSLCnano liquid chromatography system coupled to a tribrid quadrupole-ion trap-Orbitrap instrument (Orbitrap Eclipse, ThermoFisher Scientific, San Jose, CA, USA) equipped with a FAIMS Pro device. Peptides were directly loaded on a 50 cm C18 column (EasySpray ES903, 2 µm particle size and 75 µm ID, ThermalFisher, Scientific, San Jose, CA, USA) and separated using a 2-h LC-method at a flow rate of 300 nL/min. Data-dependent acquisition was performed in Top Speed mode with a duty cycle of 1 s per FAIMS compensation voltage of −40, −60, and −80 and the following parameters: spray voltage of 2100 V, ion transfer tube temperature of 275 °C, survey scan in the Orbitrap at a resolution of 120K at 200 *m/z*, scan range of 375–1575 *m/z*, AGC target = 4E5, and maximum ion injection time = 50 ms. Every parent scan was followed by a daughter scan using High Energy Collision (HCD) dissociation of top abundant peaks and detection in the orbitrap at 30 K resolution with the following settings: quadrupole isolation mode enabled, isolation window at 1.6 *m/z*, AGC target of 5E4 with maximum ion injection time of 35 ms, and HCD collision energy of 32%. Dynamic exclusion was set to 60 s. 

### 5.11. Protein Identification and Quantification 

Protein identification and label-free relative quantification was performed using the Mascot search engine in Proteome Discoverer 2.4 (ThermoFisher Scientific, San Jose, CA, USA) on a human UniProt/SwissProt database (https://www.uniprot.org; accessed on 20 May 2020) with the following parameters: trypsin cleavage rules, up to 2 missed cleavages, cysteine carbamidomethylation set as fixed modification, desthiobiotinylation of lysine residues, methionine oxidation, and N-terminal acetylation set as dynamic modifications. A 1% PSM and peptide false-discovery rate was applied in the Percolator node. Peptide and protein abundances were normalized by variance stabilizing normalization via the vsn package in R v3.6.0. Normalized protein and peptide abundances were compared between conditions of interest using a Student’s *t*-test with a Benjamini–Hochberg correction for multiple testing. Kinases were annotated from the Uniprot database. 

### 5.12. Annexin V Assay

Cells were seeded at 7500 cells/100 µL per well in a 96-well plate in DMEM supplemented with 10% FBS and allowed to adhere overnight at 37 °C. Full serum media was aspirated and replaced with the reduced serum media containing Annexin V dye and the appropriate treatment. At the time of treatment, all drugs were diluted in DMEM supplemented with 1% FBS and 1:200 Annexin V dye, protected from light. Phase and red channel live cell images were taken using the Incucyte SX5 every 3 h. The total red object intensity per image divided by the percent phase confluence of the same image was used to determine the proportion of Annexin V positive cells.

### 5.13. Statistical Analysis

One-way analysis of variance (ANOVA) was used to determine whether the means of doubling times (Figure 1C) or percentage of Annexin V-high cells (Figure 4B,C) were significantly different from each other. Tukey’s multiple comparisons test was used to correct for multiple comparisons. The log-rank (Mantel–Cox) test was used to detect significant differences between Kaplan–Meier survival curves (Figure 2A and Figure 5A,B). Moreover, *p*-values < 0.05 were considered statistically significant. Welch’s *t*-test was used to detect differences between IHC scoring in Figure 2B (panel E–F). All statistical tests were completed in GraphPad Prism.

## Figures and Tables

**Figure 1 ijms-25-06279-f001:**
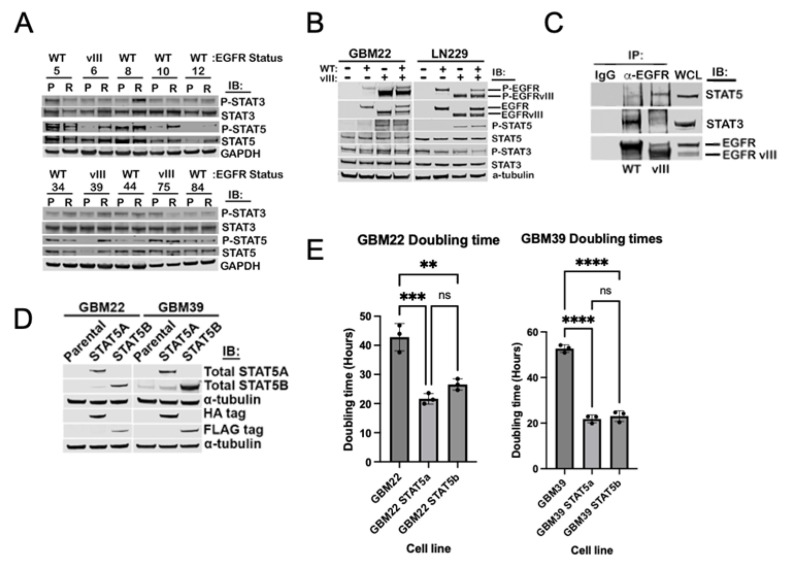
STAT5, active downstream of EGFRvIII in GBM, promotes cell proliferation in vitro. (**A**) Lysates from patient-derived xenograft tissues, either the parental tumor (P) or the induced temozolomide resistant tissue (R), were immunoblotted for phosphorylated and total STAT3 and STAT5. EGFR status is indicated above patient-derived xenograft number. (**B**) STAT5 is only phosphorylated downstream of EGFRvIII in GBM. Cell lines LN229 and GBM22 stably expressing either EGFR WT, EGFRvIII, or both constructs (EGFR/EGFRvIII) were serum starved, lysed, and immunoblotted for the indicated antibodies. (**C**) GBM22 expressing either EGFR WT or EGFRvIII was lysed and immunoprecipitated with an anti−EGFR antibody against the cytoplasmic domain (Abcam 52894; Abcam, Cambridge, UK). Precipitates were immunoblotted for total STAT5 and total STAT3. WCL whole cell lysate. (**D**) GBM22 and GBM39 stably expressing STAT5A or STAT5B were serum starved, lysed, and probed for both STAT5A/B expression and the appropriate protein tag. (**E**) Doubling time for GBM22 and GBM39 stably expressing STAT5A or STAT5B was calculated based on the average of initial cell confluence and the average of confluence at 60 h from three individual experiments measured by Incucyte SX5. **** *p* < 0.0001, *** *p* < 0.0005, ** *p* < 0.005, ns = not significant, on one way ANOVA.

**Figure 2 ijms-25-06279-f002:**
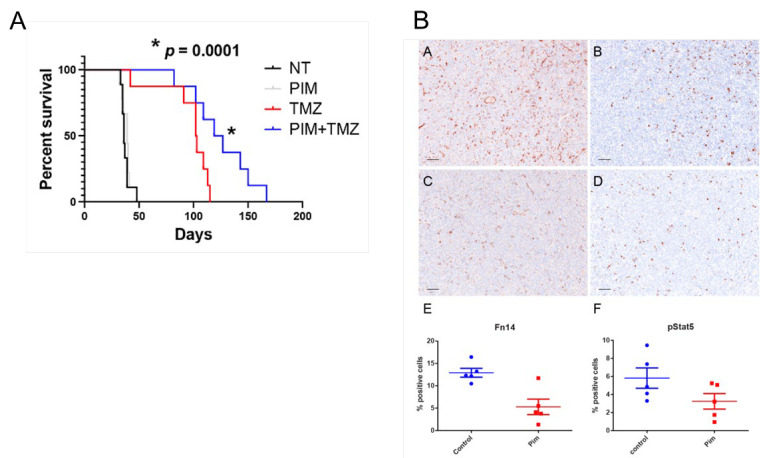
Pimozide combined with temozolomide enhances survival in orthotopic GBM6 (EGFRvIII+) tumor bearing mice compared to temozolomide alone. (**A**) Kaplan-Meier survival curve on all four treatment groups, NT = not treated (vehicle only, ORA-Plus delivered orally), PIM = 10 mg/kg pimozide, TMZ = 50 mg/kg temozolomide, PIM+TMZ = combination therapy, 10 mg/kg oral pimozide in AM, 50 mg/kg oral temozolomide in PM. Statistical significance determined by log rank test, comparing TMZ alone with PIM+TMZ group. (**B**) Representative image of immunohistochemical stain of phosphorylated STAT5 (Y694/699) and Fn14, a known STAT5 target, in the control and pimozide treatment alone to demonstrate inhibition of target. A = NT Fn14, B = NT p-STAT5, C = pimozide Fn14, D = pimozide treated p-STAT5. E and F are quantification of representative fields (Welch’s *t*-test: E—*p* < 0.01; F—*p* < 0.05). Scale bar = 100 μm).

**Figure 3 ijms-25-06279-f003:**
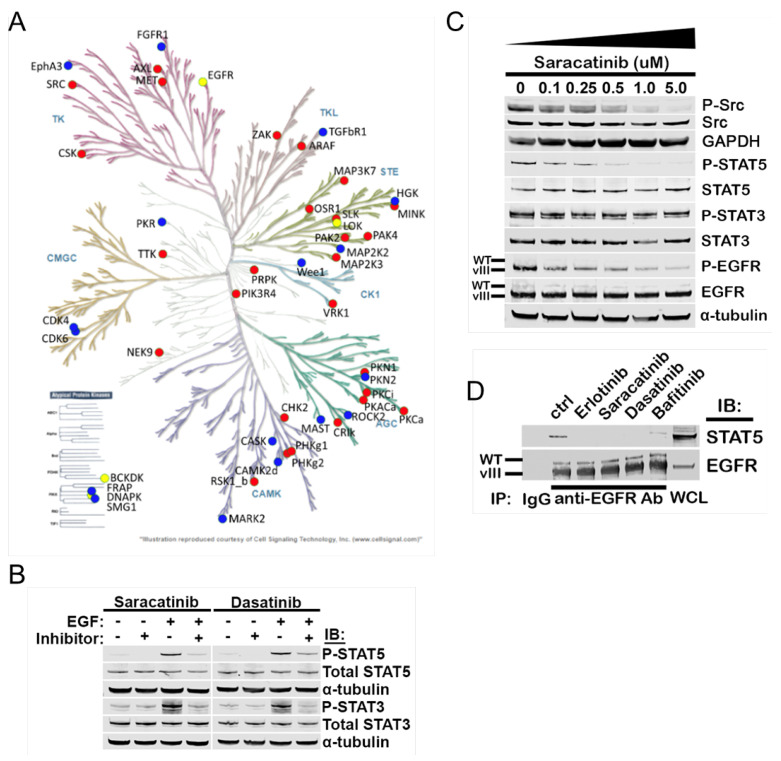
Src family kinase activity is required for EGFRvIII/STAT5 complex formation and STAT5 phosphorylation. (**A**) Summary of results of an activity−based proteome profile displayed on a kinome tree. Red dots indicate preferentially active kinase domains taking up DBT probe in EGFRvIII over EGFR WT. Blue dots are preferentially active in EGFR WT. Yellow dots indicate no difference in probe uptake between EGFRvill and EGFR WT. (**B**) Serum starved LN229 expressing EGFR/EGFRvIII were treated with 1 μM saracatinib or dasatinib for 30 min, lysed, and immunoblotted for phosphorylated and total STAT3 and STAT5. (**C**) Serum starved GBM39 cells were treated with the indicated dose of saracatinib for 30 min, lysed, and immunoblotted. (**D**) GBM39 was treated with 1 μM of the indicated drug for 30 min, with EGFR immunoprecipitated from whole cell lysates as in Figure 1C.

**Figure 4 ijms-25-06279-f004:**
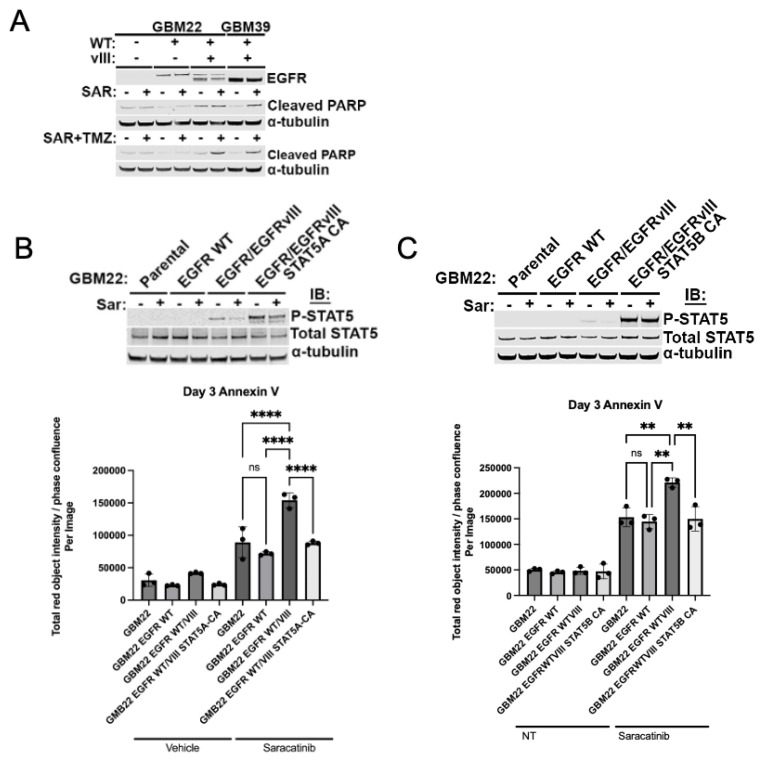
Preferential cytotoxicity of saracatinib in EGFRvIII+ cells depends on STAT5 activity. (**A**) GBM22 with indicated EGFR status and GBM39 were pre-treated with 1 μM of saracatinib 2 h prior to 24 h of combination treatment with 1 μM saracatinib and 500 μM of TMZ prior to lysate collection. (**B**) Top panel: Immunoblot validation demonstrating phosphorylated and total STAT5 in each isogenic cell line compared to constitutively active STAT5A, lanes correspond to Annexin V assay displayed in bottom panel. Cells collected for immunoblot validation were treated with 1 μM saracatinib for 30 min. Bottom panel: Annexin V dye positivity was assessed by dividing intensity (total red object intensity) by percent confluence at 3 days post treatment. Cells were treated with 1 μM saracatinib in DMEM with 1% FBS, with redosing every other day. (**C**) Immunoblot validation (top panel) and Annexin V assay (bottom panel) as in (**B**), repeated with a cell line expressing constitutively active STAT5B. ** *p* < 0.005, **** *p* < 0.00005, ns = not significant. Data points represent technical replicates within the same experiment, (**B**,**C**) represent independent experiments.

**Figure 5 ijms-25-06279-f005:**
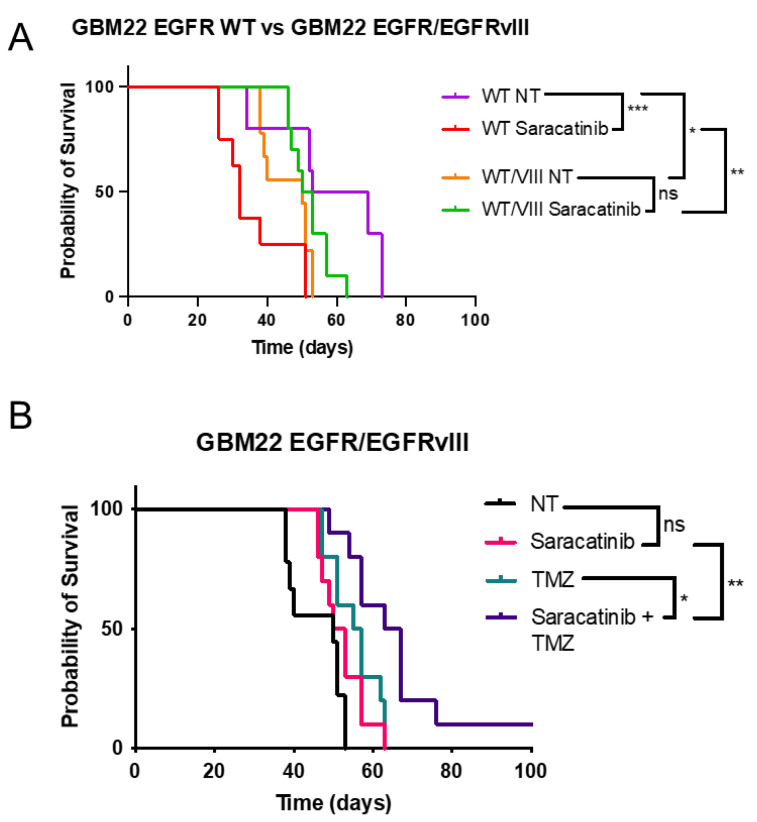
Saracatinib treatment reduces overall survival in the EGFR WT condition, but improves survival combined with TMZ in EGFRvIII+ tumor-bearing mice. (**A**) Kaplan-Meier survival curve comparing GBM22 EGFR WT and GBM22 EGFR/EGFRvIII, with and without treatment with saracatinib (25 mg/kg 5 days per week for 3 weeks). (**B**) Kaplan-Meier survival curve comparing four treatment groups in GBM22 EGFR/EGFRvIll tumors. NT = not treated (vehicle only, 50:50 ORA-Plus/ORA-Sweet delivered orally), Saracatinib 25 mg/kg, TMZ = 50 mg/kg temozolomide, saracatinib + TMZ= combination therapy, 25 mg/kg oral saracatinib in AM, 50 mg/kg oral temozolomide in PM. Statistical significance determined by log rank test. Ns = not significant, * *p* < 0.05, ** *p* < 0.005, *** *p* < 0.0005, ns = not significant.

**Table 1 ijms-25-06279-t001:** siRNA target sequences for all siRNA oligonucleotides used in this study.

siRNA	Target Sequence
Qiagen siJAK1-5	ACCGGATGAGGTTCTATTTCA
Qiagen siJAK1-6	CACGGATAACATAGCTTCAT
Qiagen siJAK2-6	CAGAATTAGCAAACCTTATAA
Qiagen siJAK2-7	AGCCATCATACGAGATCTTAA
Dharmacon siTYK2-1	GCACAAGGACCAACGUGUA
Dharmacon siTYK2-2	CAAUCUUGCUGACGUCUUG
Qiagen siSTAT5A-2	AGCGGTCGTGTTGTGAGTTA
Qiagen siSTAT5B-2	CCGAGCGAGATTGTAAACCAT
Qiagen siAXL-9	CCGGTGTTCTAAGATGTGATA
Qiagen siAXL-10	TCCAAGATTCTAGATGATTAA
Qiagen siAXL-12	CACTGTAGTTCTAAGACTCAA
Qiagen siAXL-13	AAAGTCTCTAATTCTATTAAA
Qiagen siMET-7	AACACCCATCCAGAATGTCAT
Qiagen siMET-8	ACCGAGGGAATCATCATGAAA
Qiagen siMET-9	CGCGCCGTGATGAATATCGAA
Qiagen siMET-10	CAACACCCATCCAGAATGTCA

## Data Availability

Data contained within the article.

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
