# Peer review of "EGFRvIII Confers Sensitivity to Saracatinib in a STAT5-Dependent Manner in Glioblastoma"

_ijms, 2024, doi:10.3390/ijms25116279_

Round 1

Reviewer 1 Report

Comments and Suggestions for Authors

The authors explore the role of EGFRvIII/STAT5 as a possible targetable axis in EGFRvIII GBM. The work is innovative and relevant, but needs further polishing. For instance, I would recommend relabeling results in a way that clearly indicates if a cell line is EGFRvIII mutated or wildtype, show pairwise comparison next to each other, and avoid excessive figure nesting (e.g. Figure 2). More specifically:

INTRODUCTION: the end of the introduction (lines 97 onwards) is a little wordy and more appropriate for the discussion, since results are presented and discussed. 

RESULTS: the results are too wordy. The authors include background information and methodology that belongs elsewhere in the paper. 

Figure 1D: it includes two different alpha tubulin bands, which look different. 

Figure 2.B.E-F: these graphs need quantifying. Also, if the drug is 4 times MTD, did mice exhibit any sign of toxicity (e.g. weight loss)?

Figure 3.B: alpha tubulin is again shown twice, with different blots. Has the WB been quantified? Total STAT5 and STAT3 seem decreased in the EGF + conditions. This could drive the decrease in their phosphorylated counterpart.

Figure 4.A: other pathways of cell death should be investigated besides apoptosis. 

Figure 5.B: given the dosing at toxic concentrations, could treatment itself be responsible for the worse survival observed? 

With respect to tumors that are more invasive but not larger, any invasion assay or quantification? Any increase in replicating cells or cellular density?

It would be interesting to know tumor permeation of drug, PK, PD. Further, treated and control mice should be stained for expected downstream effectors to validate the mechanism here proposed.

Lines 300-317: do the authors know if response to dasatinib was quantified based on EGFR status? It seems contradictory to propose saracatinib as a therapeutic when its analog dasatinib has failed in numerous clinical trials.

Reviewer 2 Report

Comments and Suggestions for Authors

It is an interesting review, however, I just have couple of minor queries.

1) Please add a graphical abstract.

2) Please include the conclusion section. 

3) Can authors elaborate on the imaging technique used this study. If possible, kindly include detailed scheme. 

4) Can authors include molecular docking results for better insights.

5) Add Figure 3A as an independent figure. 

6) The statement of novelty seems missing, I recommend to add at the end of the Introduction. 

7) Add discussion along with Results in the section heading. 

Round 2

Reviewer 1 Report

Comments and Suggestions for Authors

Figure 2.B.E-F: what is the statistics on this graph? Is the different statistically significant? 

Figure 3.B: The Western blot remains to be quantified given the noted differences in total Stat3 and 5. It is not clear, at this point, if there is an increase in phosphorylated analogues. 

Round 3

Reviewer 1 Report

Comments and Suggestions for Authors

We thank the authors for their revisions.